# Acute Kidney Injury and Chronic Kidney Disease Associated with a Genetic Defect: A Report of Two Cases

**DOI:** 10.3390/ijms26104681

**Published:** 2025-05-14

**Authors:** Irina Zdravkova, Eduard Tilkiyan, Hristo Ivanov, Atanas Lambrev, Violeta Dzhongarova, Gergana Kraleva, Boris Kirilov

**Affiliations:** 1Department of Propaedeutics of Internal Diseases, Medical Faculty, Medical University of Plovdiv, 4000 Plovdiv, Bulgaria; 2Nephrology Clinic, University Hospital “Kaspela”, 4000 Plovdiv, Bulgaria; eet64@yahoo.com (E.T.); nakata9721@gmail.com (A.L.); 3Second Department of Internal Diseases, Section “Nephrology”, Medical Faculty, Medical University of Plovdiv, 4000 Plovdiv, Bulgaria; 4Department of Medical Genetics, Medical Faculty, Medical University of Plovdiv, 4000 Plovdiv, Bulgaria; doctorhristoivanov@yahoo.com; 5Pediatrics Clinic, University Hospital “Sveti Georgi”, 4000 Plovdiv, Bulgaria; vili_djongarova@abv.bg; 6Department of Emergency Medicine, Internal Diseases Unit, University Hospital “Sveti Georgi”, 4000 Plovdiv, Bulgaria; kraleva.gergana16@gmail.com (G.K.); bork07@abv.bg (B.K.)

**Keywords:** gene defect, kidney damage, *SLC2A9*, *RYR1*

## Abstract

Rhabdomyolysis is one of the leading causes of acute kidney injury (AKI) and is infrequently associated with chronic kidney disease (CKD). CKD appears in diabetes mellitus and arterial hypertension, as a result of other systemic diseases and glomerulonephritis. In this study, we present two cases (one with CKD and one with AKI) that are caused by a genetic defect. A genetic examination was performed in both patients, proving that the patient with CKD has a genetic defect in the *RYR1* gene, which is observed in patients with malignant hyperthermia. Meanwhile, the patient with AKI has a homozygous pathogenic variant in *SLC2A9*, which is associated with urinary urate wasting and is characterized by asymptomatic hypouricemia and AKI after exercise. The first case is chronic rhabdomyolysis, as the patient is an athlete and performs heavy daily exercise. The second case is AKI without prior kidney damage or symptoms. Both patients did not undergo a kidney biopsy. In the first case, changes in daily routine without extreme physical exercise led to the recovery of normal kidney function. The second patient recovered from AKI without sequelae. These two cases are an example of “thinking outside the box” with respect to how genetic diseases and defects can cause kidney damage, both chronic and acute.

## 1. Introduction

Chronic kidney disease (CKD) is characterized by the presence of kidney damage or an estimated glomerular filtration rate (eGFR) of less than 60 mL/min/1.73 m^2^, persisting for 3 months or more, and the leading causes of its appearance are as follows [1,2]:Type 2 diabetes (30–50%);Type 1 diabetes (3.9%);Hypertension (27.2%);Primary glomerulonephritis (8.2%);Chronic tubulointerstitial nephritis (3.6%);Hereditary or cystic diseases (3.1%);Secondary glomerulonephritis or vasculitis (2.1%);Plasma cell dyscrasias or neoplasm (2.1%).

Sickle cell nephropathy accounts for less than 1% of End-Stage Renal Disease (ESRD) patients in the United States, while hereditary diseases are in sixth place.

When discussing hereditary chronic kidney diseases (CKDs), autosomal recessive polycystic kidney disease (ARPKD) is often one of the first conditions that comes to mind. Hereditary CKDs can be broadly categorized into monogenic and complex (multifactorial) diseases. Monogenic CKDs result from mutations in a single gene and follow Mendelian inheritance patterns (autosomal dominant, autosomal recessive, or X-linked). Recessive forms tend to manifest prenatally, in childhood, or adolescence, while dominant forms usually appear in adulthood. The genotype–phenotype correlation in dominant CKDs is often less strict, due to variable expressivity and incomplete penetrance [3]. CKDs can result from mutations affecting different renal structures, including glomerular diseases, renal cystic ciliopathies, and renal tubular disorders. Monogenic CKDs are relatively rare, while complex CKDs are more common, typically appearing later in life due to the interplay of multiple genetic and environmental factors [3].

Data from large registries indicate that the primary cause remains unresolved in 15% to 20% of patients with ESRD [4,5,6,7]. Isabel Ottlewski et al. utilized mutation analysis in patients on the kidney transplant waitlist and scrutinized the underlying renal diagnoses of 142 patients in a single-center KT waitlist. The cohort was stratified into 85 cases of determined and 57 cases of undetermined ESRD. Patients were analyzed using a renal gene panel that tests for mutations in 209 genes associated with ESRD. The most likely genetic diagnoses in 12% of the tested individuals with undetermined ESRD were established. All of these patients showed mutations in genes encoding components of the glomerular filtration barrier. Taken together, hereditary nephropathies, including autosomal dominant polycystic kidney disease, were identified in 35 of the 142 patients in the waitlist cohort. This study demonstrates the beneficial use of genetic diagnostics in significantly understanding undetermined ESRD cases prior to kidney transplantation. Interestingly, their study only yielded mutations in genes encoding glomerular, notably podocytic, structures. Despite these genes typically being associated with syndromic diseases such as Charcot–Marie–Tooth in *inverted formin-2 (INF2)*, papillorenal syndrome in *paired box 2 (PAX2)*, Denys–Drash syndrome in *Wilms tumor suppression gene (WT1)*, and Alport syndrome (AS) in the *collagen type IV alpha 5 chain (COL4A5)*, all their patients presented with isolated renal manifestations [4]. Fujimaru et al. acquired data from 1,164 patients who underwent dialysis in four different clinics in the Kanagawa Prefecture during November 2019 [8]. In this multicenter cohort, the researchers filtered out adults who were over 50 years old. They then filtered out patients who had an apparent cause for CKD, leaving 90 adults with CKD of unknown origin who had consented to genetic testing. The results revealed that 10 of the 90 patients (11% of the final cohort) had pathogenic variants in CKD-causing genes. It is particularly noteworthy that some of the hereditary renal diseases considered in this study, such as Fabry’s disease and Alport syndrome, could be diagnosed and treated early on to slow down or halt the progression of CKD [8].

There has been a drastic increase in the incidence and prevalence of CKD of unknown etiology (CKDu) across different geographical regions, without any known risk factors [9,10,11,12].

Acute kidney injury (AKI) or acute renal failure (ARF) denotes a sudden and often reversible reduction in kidney function, as measured by the glomerular filtration rate (GFR) [13,14,15,16]. AKI incidence ranges from <10% to >40% across hospital-based cohorts [17,18,19], and the etiology of AKI is divided into prerenal, renal, and postrenal. Genetic factors have been proposed as potentially responsible for the susceptibility and severity of AKI, explaining why particular patients are more prone to AKI and why different patients respond to treatment differently [20,21,22,23,24,25]. As Christian Ortega-Loubon noticed, “Understanding the contribution of environmental and genetic determinants to develop a disease represents one of the most significant challenges that researchers currently face. This is because the phenotype is shaped by genomes, the environment and their interactions” [20]. Genes frequently implicated in AKI are related to Apolipoprotein E, oxidative stress genes, vasomotor regulation genes, inflammatory and anti-inflammatory genes, and others [20].

These findings emphasize the importance of genetic evaluation in cases of unexplained CKD, particularly when clinical and laboratory results suggest an underlying metabolic or hereditary component. To further illustrate this, we present two clinical instances in which genetic analysis provided crucial insights into the etiology of kidney dysfunction.

## 2. Results

### 2.1. Patient with CKD

#### 2.1.1. Medical History and Complaints at Presentation

A patient (23 years old) was admitted in March 2024 for the first time to the Nephrology Clinic and presented with high levels of creatinine and uric acid that had been sustained for one year. The patient was not aware of any previous illnesses. He stated that he did not abuse medication or drugs. There was no family burden. He practiced football three to four times a week for about 2 h.

#### 2.1.2. Results at Admission

The patient’s results at admission are presented in Appendix A. At admission, the patient’s creatinine was normal, but he had repeatedly elevated creatinine before admission; the main reason for this was that, a few days before admission, he had not received any training.

The ultrasound of the urogenital system revealed that both kidneys had a normal position and dimensions and slightly increased echogenicity of the parenchyma. There was no evidence of nephrolithiasis and drainage disorders.

#### 2.1.3. Patient Management During Hospitalization

Treatment with low-molecular-weight heparin, intravenous fluids, and a low dosage of ACE inhibitor was carried out. The patient was discharged from the hospital with an improvement in the laboratory results. Recommendations regarding food intake and daily routine were given, as well as restrictions regarding nephrotoxic medications. The laboratory test dynamics during the patient’s stay are presented in Figure 1.

#### 2.1.4. Case Approach and Results

Considering the anamnesis and the laboratory results, we concluded that the patient’s ailment was due to rhabdomyolysis after physical exercise. We accepted that chronic mild rhabdomyolysis was the cause of the chronic kidney disease (high levels of creatinine and uric acid), which is not glomerular but instead tubulointerstitial due to the reabsorption of myoglobin by the renal tubules. Due to improvement and a lack of proteinuria, we decided to perform a kidney biopsy later if the genetic testing came back negative and if elevated creatinine reappeared and persisted, which did not happen in our case. We recommended that the patient consult a clinical geneticist to identify an enzyme deficiency as a cause.

Genetic result: Sequence analysis using whole-exome sequencing identified a heterozygous missense variant, *RYR1 c.6838G>A*, *p.(Val2280Ile)*, which was classified as likely pathogenic.

### 2.2. Patient with AKI

#### 2.2.1. Medical History and Complaints at Presentation

The second patient included in this study is a 69-year-old woman who was admitted for the first time to the Nephrology Clinic at UMHAT “Kaspela” in late August 2024. The reason for admission was a debut disturbance in kidney function parameters, which had been monitored since 2017. These include a spike in creatinine levels (347 mmol/L), proteinuria (1+), and active sediment findings.

Symptoms at the time of admission included fatigue, lumbar pain, a lack of appetite, and a recent decline in diuresis to a point at which the patient had not produced any urine for two days prior to admission. Four days before admission, the patient had undergone extreme physical exertion for her age. Arterial blood at presentation was 170/90 mmHG, and she had well-defined periorbital and lower extremity edema. She had an abundance of concomitant diseases: long-term arterial hypertension, ischemic heart disease, three coronary stent placements, COPD, bronchial asthma, and long-term type II diabetes mellitus, which were treated with sulfonylureas and a combination of insulin and a GLP-1 agonist.

#### 2.2.2. Results at Admission

The results at admission are presented in Appendix A.

Ultrasound of the urogenital system: both kidneys had a normal position and dimensions and slightly increased echogenicity of the parenchyma. Evidence indicated a single concrement measuring up to 5 mm within one of the calyces of the left kidney. There were no signs of disrupted drainage in both kidneys.

#### 2.2.3. Patient Management During Hospitalization

Therapy consisted of low-molecular-weight heparin, sodium bicarbonate, loop diuretics, and the intravenous administration of fluids, as well as the treatment of the concomitant diseases—heart failure and antihypertensive medications. Urine culture was taken, but it remained sterile. During her stay at the clinic, the patient was under an intensified insulin regimen replacing her regular anti-diabetic therapy, and she was examined by a cardiologist and an endocrinologist. The latter proposed a discontinuation of sulfonylureas due to the decline in renal function and an increase in the daily administration of insulin and GLP-1 agonist units. The patient was discharged from the hospital with an improvement in laboratory results, improved clinical presentation, and a modified daily therapy. She had a normal blood pressure upon discharge and no signs of edema. Recommendations about food intake and daily routine were given, as well as a restriction on nephrotoxic medications. The laboratory results regarding dynamics during the stay are presented in Figure 2.

#### 2.2.4. Case Approach and Results

The dynamics of the clinical laboratory results are presented in Figure 2.

Considering the recent decline in urine production, high blood pressure, edema, and laboratory results, and after a literature review, we considered that the kidney injury might be a result of renal hypouricemia; therefore, genetic testing was suggested for this patient as well.

A kidney biopsy was not performed due to the good response—the complete disappearance of edema, normal blood pressure, and a reduction in creatinine and urea within reference limits one week after discharge from the hospital.

Genetic result: Sequence analysis using whole-exome sequencing identified a homozygous variant, *SLC2A9 c.224T>G*, *p.Leu75Arg*, which was classified as likely pathogenic.

## 3. Discussion

Genetic testing has emerged as a pivotal tool in the diagnosis and management of CKD, offering several benefits that enhance patient care. A systematic review and meta-analysis including 60 studies with 10,107 adults revealed a diagnostic yield of 40% from genetic testing, with the highest yield observed in cystic kidney diseases (62%) [26]. Notably, 17% of patients received a reclassification of their diagnosis post-testing, underscoring the test’s capability to refine or alter clinical diagnoses [26].

Our study highlights the significance of genetic testing in elucidating the etiology of CKD, particularly in cases where conventional diagnostic approaches fail to provide definitive answers. The identification of a heterozygous missense variant *(RYR1 c.6838G>A*, *p.(Val2280Ile))* in a patient with CKD associated with recurrent rhabdomyolysis and a homozygous pathogenic variant in *SLC2A9* in a patient with renal hypouricemia aligns with previous findings emphasizing the genetic basis of CKD subtypes.

Mutations in the *RYR1* gene, encoding the ryanodine receptor 1, have long been associated with neuromuscular disorders such as malignant hyperthermia (MH) and central core disease (CCD). However, recent studies have highlighted a potential link between *RYR1* mutations and CKD, particularly in individuals experiencing recurrent exertional rhabdomyolysis (ER). ER results from excessive muscle breakdown, leading to the release of myoglobin, which, in high concentrations, is nephrotoxic and can cause AKI through tubular obstruction and oxidative stress. Repeated episodes of myoglobinuria-induced AKI can contribute to progressive renal fibrosis, increasing the risk of CKD. Although direct causation between *RYR1* mutations and CKD remains under investigation, recent case studies have identified patients with pathogenic *RYR1* variants who developed chronic renal impairment due to recurrent rhabdomyolysis-induced AKI [27]. This underscores the importance of genetic testing in patients with unexplained rhabdomyolysis, particularly those presenting with renal dysfunction. The early identification of *RYR1* mutations in at-risk individuals may facilitate preventive strategies, such as exercise modifications and the avoidance of triggers, to mitigate kidney damage and improve long-term renal outcomes.

Ryanodine receptor-1 is a protein found on the sarcoplasmic reticulum of rhabdomyocytes and functions as a calcium channel. It interacts with and is activated by a voltage-dependent dihydropyridine receptor (L-type calcium channel) located on the sarcolemma of the transverse tubules, thus making *RYR1* practically voltage-dependent as well. Once activated, calcium ions leak through it and exit the reticulum, flooding the sarcoplasm and ensuring a rapid increase in their concentration—the essential trigger mechanism of muscle fiber contraction. Calcium is afterwards pumped back into the sarcoplasmic reticulum by a Ca^2+^-ATPase (SERCA), thus preventing the pathologic effects of prolonged exposure to high calcium concentrations outside its deposition.

Mutations of *RYR1* may cause an increase in basal calcium concentration in the sarcoplasm and critically high concentrations during physical exercise, leading to more intense fiber contractions and a hypermetabolic state of the skeletal muscle tissue. This leads to an overwhelming amount of O_2_ consumption, increased CO_2_, and lactate production. A side effect of the factors mentioned is an increase in heat production, sometimes at a rate of 1 degree Celsius every 5 min, which exceeds physiological norms [28].

The resulting condition is termed MH and leads to a plethora of pathological events, the most significant being rhabdomyolysis—a process in which the striated muscle cell ruptures, with a following efflux of intracellular enzymes, electrolytes, and myoglobin.

Myoglobin is an oxygen-binding protein containing a single heme that is capable of binding one oxygen molecule. It is produced by most vertebrate species and almost all mammals. While found in various quantities in various muscle cell types, it is mainly produced and contained within the rhabdomyocytes of striated muscle tissue. Its primary function is to store oxygen and release it in moments of hypoxemia.

Given the mentioned factors, myoglobin is not normally present in the bloodstream unless there has been an event of mechanical or biochemical stress that causes the striated muscle cell to be damaged.

Clinical symptoms of rhabdomyolysis include myalgia, weakness and swelling of the affected muscles, dyspepsia, fever, nausea, and urine with a brownish hue due to the appearance of myoglobinuria. As muscle damage develops, fluid from the circulation extravasates into the affected muscle groups and causes edema, which (provided that a sufficient amount of muscle tissue is damaged) could lead to intravascular hypovolemia, arterial hypotension, and even shock [29].

Another possible consequence of myoglobin entering the bloodstream is the development of renal damage. This could vary from AKI to CKD, depending on the severity and frequency of rhabdomyolytic episodes.

Several pathophysiological mechanisms have been proposed. The first and oldest hypothesis is that abundant quantities of myoglobin molecules are filtered through the glomeruli and enter the renal tubules. At an acidic pH, they aggregate and bind to uromodulin, which leads to the formation of intratubular casts. These then cause an obstruction of the nephron, thus hindering further glomerular filtration and the production of primary urine [30].

Other mechanisms include an increase in oxidative stress, which is directly or indirectly caused by the released heme of the myoglobin.

Once in the tubules, the cellular uptake of myoglobin is initiated by the endocytic receptors megalin and cubilin, which promote endocytosis [29]. Within the tubule cell, the heme is broken down by hemoxygenase and ferrochelatase to carbon monoxide, biliverdin, and a free ferrous ion. The latter then induces the production of O_2_ radicals and a ferric ion through the Fenton reaction [31]. The Fenton reaction is presented in Equation (1).

The Fenton reaction: H_2_O_2_ + Fe^2+^→ OH^−^ +Fe ^3+^(1)

The emerging reactive species oxidize some of the myoglobin into metmyoglobin and then into ferrylmyoglobin, with lipid peroxidation occurring at each step of the process. Furthermore, redox cycling occurs between the ferric and ferryl forms of myoglobin, which leads to the significant production of O-radicals [29]. The peroxidized lipids lead to damage to mitochondrial membranes, disrupt their transporter molecules, and uncouple the respiratory chains, resulting in ATP depletion, cellular damage, and death.

Additionally, high concentrations of ROS decrease the production of NO in various ways, which leads to renal vasoconstriction and ischemia [32]. The pathophysiological mechanisms in myoglobin-induced tubule damage are presented in Figure 3.

Uric acid is an end product of purine metabolism in humans. Having lost the ability to synthesize active uricase, in the course of our phylogenesis, we rely exclusively on the renal excretion of produced uric acid in order to maintain its serum levels within normal ranges. Uric acid seems to have the capacity to act both as an intracellular pro-oxidant and as a potent and main extracellular antioxidant. Due to the latter quality, its molecule is still of value to organisms [33]. Therefore, during the course of evolution, complex mechanisms of renal reabsorption have also developed in order to prevent hypouricemia and the oxidative stress to which it leads. The transporters with the highest capacity—and, thus, the greatest relevance to this study—are underlined in Figure 4.

One such mechanism is mediated by the product of the *SLC2A9* gene. *SLC2A9* is found within chromosome 4 and encodes a certain type of glucose transporter—solute carrier family 2, facilitated glucose transporter member 9 (GLUT 9), which is found in the cells of the kidney, cartilage, liver, and the placenta. Two isoforms have been identified—a long isoform and a short isoform. The first is expressed chiefly on the basolateral membranes of the nephron’s proximal tubules and hepatocytes, while the second isoform is located only on the luminal membranes of tubule cells and within the placenta [34]. *SLC2A9* has been shown to have a high affinity toward glucose. However, this is not the sole metabolite that it transports. Fructose and, more recently (as well as more strongly), uric acid have also been found to be transferred by it. It is one of the three main proteins responsible for luminal uric acid reabsorption within the proximal tubules, along with URAT1 and OAT4, while ABCG2 is the main excreting transporter [34]. SLC2A9 is the only basolateral exporter, which brings uric acid out of the tubular cell and back into the interstitium and blood circulation [35].

Multiple single-nucleotide polymorphisms within the encoding gene have been associated with patients developing hyperuricemia and gout, while rare cases present with hypouricemia and excessive renal loss of uric acid, with resulting hyperuricosuria [34].

In these latter cases, the appearing condition is termed renal hypouricemia type 2 (RHUC2), while renal hypouricemia type 1 (RHUC1) is caused by mutations in URAT1.

Clinical presentation in both types consists of abdominal pain, hematuria, nephrolithiasis, myalgia, and AKI, oftentimes after physical exercise, making it a kind of exercise-induced acute renal failure [35].

Several pathophysiological mechanisms have been proposed:One is that exercise causes an increase in uric acid production and a spike in serum levels. Given that the latter already has abnormally high excretion, its tubular concentration increases even more, which leads to the precipitation and deposition of crystals within the tubules. Due to this, nephrons may become obstructed, and the production of primary urine is impaired [35].Another is that extraordinarily high tubular concentrations of uric acid have been shown to activate Toll-Like Receptor 4 (TLR-4) on the apical surface of renal cells, which then leads to the activation of the NLRP3 inflammasome complex. This results in the induced production of interleukin 1β (IL-1β), which then acts as an activator of sympathetic fibers. The result of this is the vasoconstriction of glomerular arterioles, which further decreases filtration and interstitial blood flow [36].Mutations in GLUT9 lead to a partial (yet substantial) decrease in luminal urate reuptake because other apical transporters continue to function. The basolateral efflux of uric acid, however, is severely diminished due to GLUT9 being the sole basolateral exporter. Therefore, there is an increase in the uric acid concentration within tubular cells, which leads to the production of ROS and oxidative stress (due to its property of being an intracellular oxidant) [37].A fourth mechanism is that during physical exercise, the body is subjected to elevated levels of oxidative stress, which, in the absence of sufficient amounts of antioxidants (i.e., uric acid), leads to oxidative damage, renal vasoconstriction, and ischemia, resulting in tubular cell damage (acute tubular damage) [35].

Histological results from kidney biopsies tend to indicate that intratubular precipitation and nephron obstruction are a relatively rare mechanism, with most patients lacking signs of massive urate crystal deposits in their tubules [35]. The pathophysiological mechanisms of kidney damage in renal hypouricemia type 2 are presented in Figure 5.

Mutations in the *SLC2A9* gene, encoding the GLUT9 transporter, play a pivotal role in renal uric acid handling. These mutations impair GLUT9 function, leading to increased urinary uric acid excretion (hyperuricosuria) and a predisposition to exercise-induced acute kidney injury (AKI) due to urate crystallization in renal tubules [38]. While patients with SLC2A9-associated renal hypouricemia (RHUC) often remain asymptomatic, they may develop urolithiasis, nephrolithiasis, and AKI, particularly after intense physical exertion [39].

The homozygous *SLC2A9 c.224T>G*, *p.Leu75Arg* variant, identified in a 71-year-old patient with hypouricemia and AKI, is consistent with renal hypouricemia type 2 (RHUC2). This disorder results in impaired urate reabsorption, leading to excessive uric acid excretion (hyperuricosuria) and a predisposition to AKI due to urate crystal deposition in renal tubules [37]. While RHUC could remain asymptomatic, cases like this demonstrate its potential to cause significant renal injury, particularly in individuals with comorbid conditions such as diabetes, COPD, and cardiovascular disease, which may exacerbate renal vulnerability.

The results of our study emphasize the importance of incorporating genetic testing into the diagnostic workup of CKD, particularly in patients without clear risk factors. The benefits of genetic testing in CKD include the following:

Refinement of diagnosis: genetic findings allow for a more precise classification of CKD subtypes, avoiding misdiagnosis and inappropriate treatment.

Personalized medicine approaches: identifying specific mutations enables targeted interventions, such as dietary modifications, pharmacological adjustments, and the avoidance of nephrotoxic medications or activities.

Familial screening and counseling: inherited kidney disorders can be detected in asymptomatic family members, allowing for early monitoring and preventative strategies.

### Future Research Directions

Given the increasing recognition of genetic factors in CKD, future research should focus on several key areas. Expanding genetic testing programs in routine nephrology clinics can significantly improve early diagnosis and disease management. Studies have shown that genetic testing enhances diagnostic outcomes and provides critical insights into the genetic basis of kidney diseases [40,41]. Another critical area of research involves the functional characterization of novel mutations that are discovered through genomic sequencing. While next-generation sequencing has uncovered a multitude of genetic variants associated with CKD, many of these variants remain of uncertain significance. To better understand their role in disease development, future studies must investigate the functional impacts of these mutations on renal physiology. This could involve using in vitro models, animal studies, or patient-derived organoids to assess how these genetic variants influence kidney function at the molecular and cellular levels. Through determining whether these mutations affect key processes such as ion transport, glomerular filtration, or tubular function, researchers can better understand their contribution to the pathogenesis of CKD. Functional characterization of these mutations will not only clarify their clinical significance but could also provide insights into potential therapeutic targets, paving the way for more effective treatments.

## 4. Materials and Methods

### 4.1. Materials

In both patients, the diagnosis was confirmed by laboratory tests and genetic testing.

### 4.2. Genetic Testing

Whole-exome sequencing was performed, followed by bioinformatics and quality control. The pathogenicity potential of the identified variants was assessed by considering the predicted consequence, the biochemical properties of the codon change, the degree of evolutionary conservation, and a number of reference population databases and mutation databases, such as (but not limited to) the 1000 Genomes Project, gnomAD, ClinVar, and HGMD Professional. Variant classification was based on the ACMG guideline 2015.

#### 4.2.1. Patient with CKD

Laboratory process: The total genomic DNA was extracted from the biological sample using the bead-based method. The quantity of DNA was assessed using the fluorometric method. After the assessment of DNA quantity, the qualified genomic DNA sample was randomly fragmented using non-contact, isothermal sonochemistry processing. The sequencing library was prepared by ligating sequencing adapters to both ends of DNA fragments. Sequencing libraries were size-selected with the bead-based method to ensure an optimal template size and amplified using the polymerase chain reaction (PCR). Regions of interest (exons and intronic targets) were targeted using the hybridization-based target capture method. The quality of the completed sequencing library was controlled by ensuring the correct template size and quantity to eliminate the presence of leftover primers and adapter–adapter dimers. Ready sequencing libraries that passed the quality control were sequenced using Illumina’s sequencing-by-synthesis method using paired-end sequencing (2 × 150 bases). Primary data analysis, converting images into base calls and associated quality scores, was carried out using the sequencing instrument using Illumina’s proprietary software (https://www.illumina.com/products/by-type/informatics-products.html, accessed on 5 May 2025), generating CBCL files as the final output.

Bioinformatics and quality control: Base-called raw sequencing data were transformed into FASTQ format using Illumina’s software (bcl2fastq). The sequence reads of each sample were mapped to the human reference genome (GRCh37/hg19). The Burrows–Wheeler Aligner (BWA-MEM) software (https://bio-bwa.sourceforge.net/, accessed on 5 May 2025) was used for read alignment. Duplicate read marking, local realignment around indels, base quality score recalibration, and variant calling were performed using GATK algorithms (Sentieon) for nDNA. Variant data were annotated using a collection of tools (VcfAnno and VEP) with a variety of public variant databases, including but not limited to gnomAD, ClinVar, and HGMD. The median sequencing depth and coverage across the target regions for the tested sample were calculated based on MQ0 aligned reads. The sequencing run included in-process reference sample(s) for quality control, which passed our thresholds for sensitivity and specificity. The patient’s sample was subjected to thorough quality control measures, including assessments for contamination and sample mix-up. Copy number variations (CNVs), defined as single-exon or larger deletions or duplications (Del/Dups), were detected from the sequence analysis data using a proprietary bioinformatics pipeline. The difference between the observed and expected sequencing depth at the targeted genomic regions was calculated, and regions were divided into segments with the variable DNA copy number. The expected sequencing depth was obtained by using other samples processed in the same sequence analysis as a guiding reference. The sequence data were adjusted to account for the effects of varying guanine and cytosine content.

Interpretation: The pathogenicity potential of the identified variants was assessed by considering the predicted consequence; the biochemical properties of the codon change; the degree of evolutionary conservation; and a number of reference population databases and mutation databases, such as (but not limited to) the 1000 Genomes Project, gnomAD, ClinVar, and HGMD Professional. For missense variants, in silico variant prediction tools such as SIFT, PolyPhen, and MutationTaster were used to assist with variant classification. In addition, the clinical relevance of any identified CNVs was evaluated by reviewing the relevant literature and databases such as the 1000 Genomes Project, Database of Genomic Variants, ExAC, gnomAD, and DECIPHER. For interpretation of mtDNA variants, specific databases, including, e.g., Mitomap, HmtVar, and 1000G, were used. The clinical evaluation team assessed the pathogenicity of the identified variants by evaluating the information in the patient referral, reviewing the relevant literature, and manually inspecting the sequencing data if needed. Reporting was carried out using HGNC-approved gene nomenclature and mutation nomenclature while following the HGVS guidelines. Likely benign and benign variants were not reported.

Variant classification: Variant classification followed the Blueprint Genetics Variant Classification Schemes, which were modified in the ACMG guideline 2015. Minor modifications were made to increase the reproducibility of the variant classification and improve the clinical validity of the report. The classification and interpretation of the variant(s) identified reflect the current state of Blueprint Genetics’ understanding at the time of this report. Variant classification and interpretation are subject to professional judgment and may change for a variety of reasons, including but not limited to updates in classification guidelines and the availability of additional scientific and clinical information.

Confirmation of sequence alterations: Sequence variants classified as pathogenic, likely pathogenic, and variants of uncertain significance (VUS) were confirmed using bi-directional Sanger sequencing when they did not meet our stringent NGS quality metrics for a true positive call.

Confirmation of copy number variants: CNVs (deletions/duplications) were confirmed using a digital PCR assay if they covered fewer than 10 exons (heterozygous), fewer than 3 exons (homo/hemizygous), or were not confirmed at least 3 times previously at laboratory. Furthermore, CNVs of any size were not confirmed when the breakpoints of the call could be determined.

Analytic validation: The laboratory-developed test was independently validated by Blueprint Genetics. The validated performance of this whole-exome sequencing laboratory assay is as follows: 99.6% sensitivity for SNVs, 97.6% for 2–50 bps indels, 100% for one-exon deletion, and 82% for 1–10 exon duplications. Specificity is >99.9% for most variant types. It does not detect very low-level mosaicism, as a variant with a minor allele fraction of 14.6% can be detected in 90% of the cases. Detection performance for mtDNA variants (analytic and clinical validation) is as follows: sensitivity for SNVs and INDELs is 100.0% (5–100% heteroplasmy level), and for simulated gross deletions (500–5000 bp), it is >99.9%. Specificity is >99.9% for all.

Regulation and accreditations: The test was developed, and its performance characteristics were determined by Blueprint Genetics (see the analytical validation). It has not been cleared or approved by the US Food and Drug Administration. This analysis was performed in a CLIA-certified laboratory (#99D2092375) accredited by the College of American Pathologists (CAP #9257331) and the FINAS Finnish Accreditation Service (laboratory No. T292), accreditation requirement SFS-EN ISO 15189:2013 [42]. All the tests are under the scope of the ISO 15189 accreditation, excluding mitochondrial DNA testing, which was performed on a whole-exome sequencing assay. PERFORMING SITE: BLUEPRINT GENETICS OY, KEILARANTA 16 A-B, 02150 ES-POO, FINLAND.

#### 4.2.2. Patient with AKI

Genomic DNA was extracted from an EDTA blood specimen using a standard protocol. Exome capture was performed using xGen Exome Research Panel v2 supplemented with xGen human mtDNA panel and xGen Custom Hyb Panel v1 (Integrated DNA Technologies, Coralville, IA, USA). Sequencing was performed using NovaSeq X (Illumina, San Diego, CA, USA). In total, 11,719,064,293 bases of sequence were generated and uniquely aligned to the Genome Reference Consortium Human Build 38 (GRCh38) and Revised Cambridge Reference Sequence (rCRS) of the mitochondrial genome, generating a 172.41 mean depth of coverage within the 34,212,647 bases of the captured region, which is approximately 99.3% of the RefSeq protein coding region. Approximately 99.50% of the targeted bases were covered to a depth of ≥20×. Despite the insufficient coverage across 0.50% of the bases (see below for details), these metrics were consistent with high-quality exome sequencing data and were deemed adequate for analysis. In total, 65,603 single-nucleotide variants (SNV) and 12,019 small insertions and deletions (INDEL) were identified. Sequencing data analysis and variant interpretation were performed using 3 billion’s proprietary system, EVIDENCE v4.2 (Clin Genet. 2020;98:562–570). EVIDENCE incorporates a bioinformatics pipeline for calling SNV/INDEL based on the GATK best practices (GATK v4.4.0, Genome Res. 2010;20:1297–303) and Manta v1.6.0 (Bioinformatics. 2016;32:1220–2) for calling CNV (copy number variants) based on paired-end information, as well as 3bCNV v2.1, an internally developed tool, for calling CNV (copy number variants), including aneuploidy based on the DOC information. It also incorporates Mutect2 v4.4.0 (Genome Res. 2010;20:1297–303) for calling lower-level heteroplasmic SNV/INDEL in the mitochondrial genome, ExpansionHunter v5.0.0 (Bioinformatics. 2019;35:4754–6) for calling repeat expansion variants, MELT v2.2.2 (Genome Res. 2017;27:1916–29) for calling mobile element insertion variants, and AutoMap v1.2 (Nat Commun. 2021;12:518) for detecting regions of homozygosity (ROH). Variant Effect Predictor v104.2 (VEP, Ensembl, Genome Biology 2016;17:122) was used for variant annotation. Variants were prioritized based on the guideline recommended by the American College of Medical Genetics and Genomics (ACMG) and the Association for Molecular Pathology (AMP) (Genet Med. 2015;17:405–424, Genet Med. 2020:22:245–257, and Hum Mutat. 2020;41:2028–2057) in the context of the patient’s phenotype, relevant family history, and previous test results provided by the ordering physician. Only variants deemed clinically significant and relevant to the patient’s clinical indications at the time of variant interpretation are reported. Based on internal studies validating the accuracy of the variants called with high quality scores, only low-quality variants are confirmed by Sanger sequencing.

This test is developed by 3billion for the purpose of identifying single-nucleotide variants (SNV), small insertions/deletions (INDEL, <50 bp), large (≥3 consecutive exons) copy number variants, mobile element insertion variants, and repeat expansion variants within the targeted genomic regions. Repeat expansion detection is possible for the following 18 genes. Repeat expansion number may be underestimated for the starred (*) gene with compromised sensitivity (AR, ARX, ATN1, ATXN1, ATXN2, ATXN3, ATXN7, ATXN8OS*, CACNA1A, COMP, FOXL2, HOXD13, HTT, PABPN1, PHOX2B, PRDM12, TBP, and ZIC2). Only SNV/INDEL (>10% heteroplasmic level) are called within the mitochondrial genome. The laboratory is certified under the College of American Pathologists (CAP#:8750906) and Clinical Laboratory Improvement Amendments (CLIA#: 99D2274041) as qualified to perform high-complexity clinical laboratory testing. Assay validation and clinical validation were performed while following the Korea Institute of Genetic Testing Evaluation and the American College of Medical Genetics and Genomics (ACMG) Technical Standards and Guidelines Section G (https://www.acmg.net/PDFLibrary/Standards-Guidelines-Clinical-Molecular-Genetics.pdf accessed on 5 May 2025). Accreditations and Certifications: CAP License #8750906, AU-ID# 2052626, and CLIA ID #99D2274041.

## 5. Conclusions

Our findings reinforce the importance of genetic testing in CKD diagnostics and patient management. The identification of pathogenic variants in RYR1 and SLC2A9 in our patients illustrates how genetic testing can uncover previously unrecognized disease mechanisms and inform clinical decision making. While challenges remain regarding cost, accessibility, and interpretation, integrating genetic testing into nephrology practice holds great promise for personalized treatment, early intervention, and improved patient outcomes. Future research should focus on the expansion of genetic testing, the functional characterization of novel variants, and the development of targeted therapies to revolutionize CKD management.

## Figures and Tables

**Figure 1 ijms-26-04681-f001:**
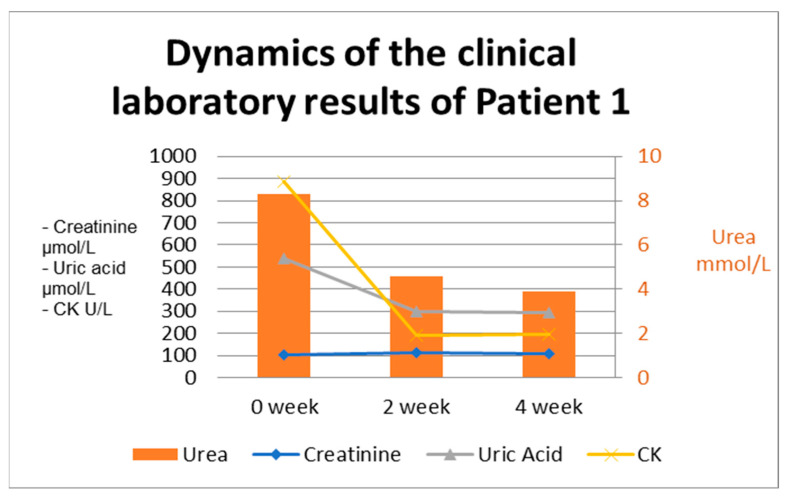
The dynamics of the clinical laboratory results in Patient 1. CK—creatine kinase.

**Figure 2 ijms-26-04681-f002:**
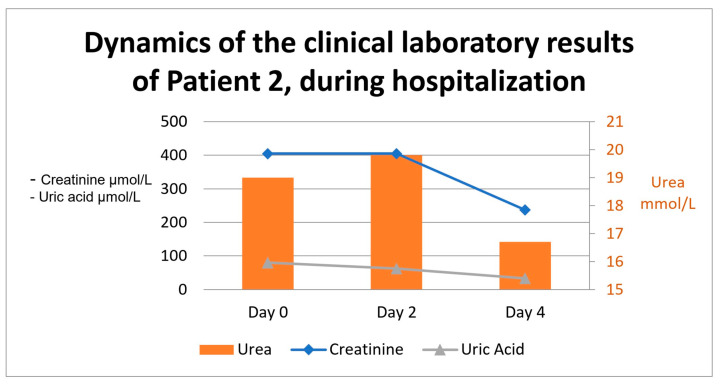
The dynamics of the clinical laboratory results in Patient 2.

**Figure 3 ijms-26-04681-f003:**
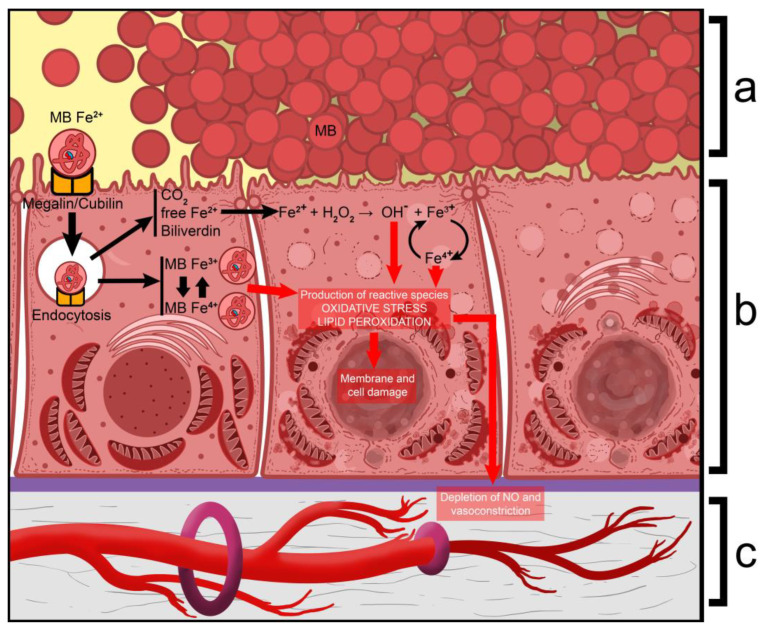
The pathophysiological mechanisms in myoglobin-induced tubule damage. (**a**)—casts, consisting of myoglobin, obstructing the tubular lumen, and causing blockage of the nephron; (**b**)—cellular damage, caused by mechanisms that lead to the production of ROS, lipid peroxidation, and cellular damage; (**c**)—vasoconstriction of interstitial blood vessels. Fe^2+^—ferrous iron; Fe^3+^—ferric iron; Fe^4+^—ferryl iron; MB/MB Fe^2+^—myoglobin; MB Fe^3+^—ferrimyoglobin/metmyoglobin; MB Fe^4+^—ferrylmyoglobin; NO—nitric oxide.

**Figure 4 ijms-26-04681-f004:**
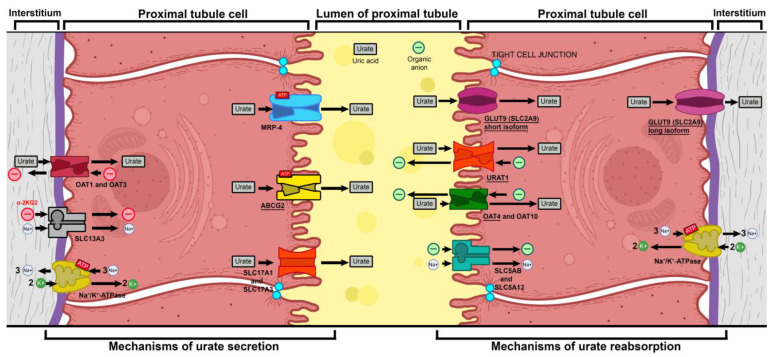
A schematic representation of uric acid (urate) transportation within the proximal tubule: GLUT9 (SLC2A9)—Glucose Transporter 9 (Solute Carrier 2A9)—a high-capacity transporter with a high affinity for glucose and uric acid and low affinity for fructose; URAT1—Urate Transporter 1—a urate/organic anion antiporter; OAT1—Organic anion transporter 1—a urate/α-2ketoglutarate 2^−^ antiporter; OAT3—Organic anion transporter 3—a urate/α-2ketoglutarate 2^−^ antiporter; OAT4—Organic anion transporter 4—a urate/organic anion antiporter; OAT10—Organic anion transporter 10—a urate/organic anion antiporter; SLC5AB—Solute Carrier 5AB—a sodium/organic anion cotransporter; SLC5A12—Solute Carrier 5A12—a sodium/organic anion cotransporter; SLC13A3—Solute Carrier 13A3—a sodium/α-2ketoglutarate 2^−^ cotransporter; SLC17A1—Solute Carrier 17A1—a transporter of uric acid; SLC17A3—Solute Carrier 17A3—a transporter of uric acid; MRP1—Multi-drug Resistance Protein 1—an active transport protein with an affinity for a wide variety of molecules, including urate; ABCG2—ATP-binding cassette superfamily G member 2—an active transport protein with a high capacity to transport urate; Na^+^/K^+^-ATPase—a broadly expressed sodium/potassium active transport antiporter, which brings 2 potassium ions into the cell and exports 3 sodium ions; α-2KG^2^—α-ketoglutarate; -NB! Some transporters are illustrated as sharing a single model due to their structural and functional similarity. These include OAT1 and OAT3, OAT4 and OAT10, SLC5AB and SLCA5A12, and SLC17A1 and SLC17A3.

**Figure 5 ijms-26-04681-f005:**
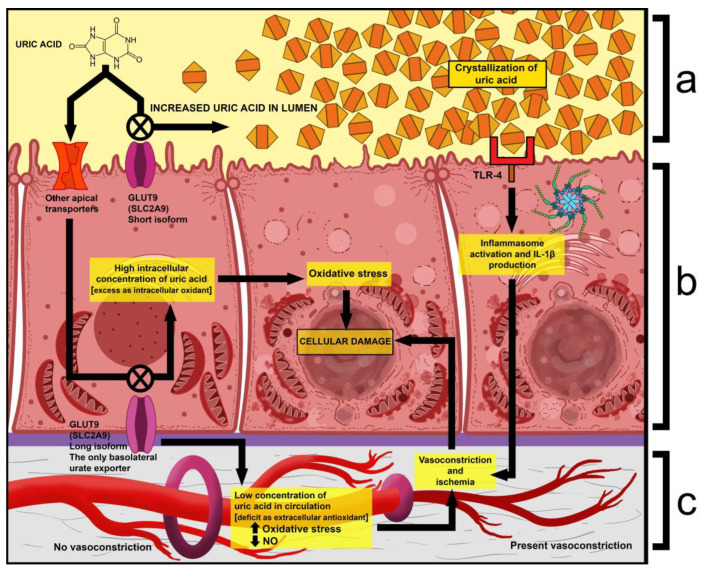
The pathophysiological mechanisms of kidney damage in renal hypouricemia type 2. (**a**)—the crystallization of uric acid within the tubular lumen and nephron obstruction; (**b**)—the mechanisms that play out at the level of renal tubule cells, where there is an excess amount of uric acid, acting as an oxidant; (**c**)—the vasoconstriction of interstitial blood vessels, where there is a deficiency of uric acid, which acts as an antioxidant; TLR-4—Toll-Like Receptor 4; IL-1β—interleukin-1β; NO—nitric oxide.

## Data Availability

The data presented in this study are available from the corresponding author upon request. The data are not publicly available due to national legal restrictions.

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
