# Peer review of "Acute Kidney Injury and Chronic Kidney Disease Associated with a Genetic Defect: A Report of Two Cases"

_ijms, 2025, doi:10.3390/ijms26104681_

Round 1

Reviewer 1 Report

Comments and Suggestions for Authors

The paper Acute Kidney Injury and Chronic Kidney Disease, Associated with Genetic Defect: A Report of Two Cases  is a case report that illustrates how genetic testing can uncover previously unrecognized disease mechanisms and inform clinical decision-making. The manuscript is well written and organized, provides all background to the reader less familiar with the subject, thereby including all necessary international state-of-the-art. The conclusions are valid and fully supported by the results.The cited references are relevant and mostly from recent publications.

Minor changes should be done: Line 420 and 421 should be deleted. The paper can be accept after minor revision.

Author Response

Comments 1: The paper Acute Kidney Injury and Chronic Kidney Disease, Associated with Genetic Defect: A Report of Two Cases is a case report that illustrates how genetic testing can uncover previously unrecognized disease mechanisms and inform clinical decision-making. The manuscript is well written and organized, provides all background to the reader less familiar with the subject, thereby including all necessary international state-of-the-art. The conclusions are valid and fully supported by the results. The cited references are relevant and mostly from recent publications.

Response 1: Thank you very much for the kind words and appreciation of our work.

Comments 2: Minor changes should be done: Line 420 and 421 should be deleted. The paper can be accept after minor revision.

Response 2: Line 420 and 421 are deleted.

Thank you for your review, it is important for if you find it interesting and relevant and thank you for your comments!

Reviewer 2 Report

Comments and Suggestions for Authors

In this manuscript, Zdravkova and colleagues report two cases of acute kidney injury and chronic kidney disease, associated with a genetic defect.

Major:

  1. It's unclear if the authors believe that the patient didn't take drugs or had no previous illness because they wrote, "He denies any previous illnesses. Denies using any medications or drugs."

If the attending medical personnel suspected drug usage, a drug test should have been ordered. Otherwise, the authors should change the language. Instead of claiming that the patient "denies any previous illnesses", it would be better to say "The patient isn't aware of any previous illnesses". The patient also doesn't "deny using any medications or drugs", but "states that he is not abusing medication or drugs".

  1. To authors measured CK at level of 887 U/L and diagnosed rhabdomyolysis. The clinical definition for mild CK is 1000 - 5000U/L. It would be necessary to determine the myoglobulin level to determine that rhabdomyolysis was present.

Minor:

- In Table S1 and S2 the authors state that the normal range of bilirubinuria is "not elevated". The authors should replace this with "negative or trace amounts".

- In Table S1 the authors state that the normal range of proteinuria is 1600ml/24h, and in S2 they wrote "1200ml/24h". Please correct!

- Please carefully edit this manuscript for proper language, grammar and syntax.

- line 420 is written in Cyrillic and could not be reviewed.

Author Response

Comments 1: It's unclear if the authors believe that the patient didn't take drugs or had no previous illness because they wrote, "He denies any previous illnesses. Denies using any medications or drugs."

If the attending medical personnel suspected drug usage, a drug test should have been ordered. Otherwise, the authors should change the language. Instead of claiming that the patient "denies any previous illnesses", it would be better to say "The patient isn't aware of any previous illnesses". The patient also doesn't "deny using any medications or drugs", but "states that he is not abusing medication or drugs".

Response 1: Corrected.

Comments 2: To authors measured CK at level of 887 U/L and diagnosed rhabdomyolysis. The clinical definition for mild CK is 1000 - 5000U/L. It would be necessary to determine the myoglobulin level to determine that rhabdomyolysis was present.

Response 2: The patient had no training a few days before admission and still his CK was 887, we believe that after training it exceeds those 113 U/l, even more. We did not measure his myoglobulin than, now there is no point, since he is not training anymore and his creatinine as well as CK went down to normal.

Comments 3: In Table S1 and S2 the authors state that the normal range of bilirubinuria is "not elevated". The authors should replace this with "negative or trace amounts".

Response 3: Corrected.

Comments 4: In Table S1 the authors state that the normal range of proteinuria is 1600ml/24h, and in S2 they wrote "1200ml/24h". Please correct!

Response 4: Corrected.

Comments 5: Please carefully edit this manuscript for proper language, grammar and syntax.

Response 5: The manuscript underwent English Editing.

Comments 6: line 420 is written in Cyrillic and could not be reviewed.

Response 6: Deleted.

4. Response to Comments on the Quality of English Language

Point 1:

Response 1:    (in red)

5. Additional clarifications

Thank you for your review, it is important for us that you find it interesting and relevant and thank you for your comments, they really made the article better!

Reviewer 3 Report

Comments and Suggestions for Authors

In this manuscript the Authors provided a description of two cases report of AKI and CKD associate to genetic defect. It is not clear if the reported genetic defects are associated with rhabdomyolysis, it could be better explained. The Authors could improve results section by adding the reasons why the investigations were or were not carried out (eg. why kidney biopsies were not performed?) or why some tratments were used rather than ones. In figure 2 the CK detection is missing. Please add footnotes for figure 1 and 2 to explain abbreviations. 

Author Response

Comments 1: In this manuscript the Authors provided a description of two cases report of AKI and CKD associate to genetic defect. It is not clear if the reported genetic defects are associated with rhabdomyolysis, it could be better explained.

Response 1: We believe that chronic mild rhabdomyolysis is the reason for kidney damage in the first case, and in second hypouricemia is reason for AKI. It will be underlined in 2.1.4. Case approach and results and 2.2.4. Case approach and results.

Comments 2: The Authors could improve results section by adding the reasons why the investigations were or were not carried out (eg. why kidney biopsies were not performed?) or why some tratments were used rather than ones.

Response 2: It will be underlined in 2.1.4. Case approach and results as well as in 2.2.4. Case approach and results why did not performed kidney biopsy. Treatment is not the subject of our study, as well it is not the accent of the current special issue, this is the reason why it was mentioned but not explained in details.

Comments 3: In figure 2 the CK detection is missing. Please add footnotes for figure 1 and 2 to explain abbreviations.

Response 3:  CK in figure 2 was left by mistake, it will be corrected since in second case the hypouricemia is the reason for AKI.

5. Additional clarifications

Thank you for your review, it is important for us if you find it interesting and relevant and thank you for your comments!

Round 2

Reviewer 2 Report

Comments and Suggestions for Authors

The authors made appropriate changes to the manuscript and significantly edited the language. The manuscript is recommended for publication.

Author Response

Thank you!